# Failure Localization for Enterprise Text-to-SQL Agents: Lessons from a Production-Representative Financial Case Study

Heeyong Eun[1]

KPMG Korea, Seoul, South Korea
`heeyonge@gmail.com`

**Abstract.** Large-language-model Text-to-SQL agents are attractive AI-assisted software engineering tools, yet enterprise deployments often fail *silently*: generated SQL can execute while answering the wrong intent due to mis-grounded terms, wrong anchors, meaningless joins, or ungrounded filters. The core challenge is diagnostic because failures can originate at different pipeline stages, but typical retrieval-and-generation systems do not preserve intermediate commitments in an inspectable form. I present an artifact-level failure-localization method that fixes major intermediate decisions as structured artifacts and compares them against reference artifacts derived from administrator-corrected SQL via a deterministic "Gold Reverser." Under paired replay on a production-representative financial metadata snapshot, the revised configuration improves schema linking and reduces structural errors compared with a baseline. The contribution is not a new Text-to-SQL model, but an operational debugging and correction workflow that improves diagnosability and correctability for enterprise-grade Text-to-SQL systems.

**Keywords:** Text-to-SQL; enterprise LLM agents; AI for software engineering; failure localization; regression testing; schema linking; structural validation

## 1 Introduction

Text-to-SQL agents translate natural-language requests into executable database queries and are increasingly deployed as AI-assisted software engineering tools that produce code-like artifacts. In enterprise settings, however, a generated SQL query can be *executable* yet still be *wrong*: it may answer a different business question, ground a term to an incorrect physical column, choose an inappropriate anchor table, construct a plausible-but-meaningless join, or introduce an unsupported filter. This gap is amplified in large, heterogeneous schema environments compared with benchmark-style scenarios such as Spider [5] and schema-linking approaches such as RAT-SQL [2].

The core operational challenge is diagnostic. Failures can originate at multiple stages—term interpretation, schema grounding, table selection and join planning,

or filter construction—but typical pipelines do not preserve intermediate commitments in an inspectable form. Decisions are absorbed into prompts or model traces and only surface in the final SQL, making it difficult to localise where the pipeline first diverged. Teams then resort to repeated prompt or rule patching, which is hard to reproduce and improve cumulatively.

I make one claim. If intermediate decisions are fixed as structured artifacts and compared against reference artifacts derived from administrator-corrected SQL, failures can be localised to the *first wrong commitment* (e.g., grounding vs. linking vs. join vs. filter), enabling systematic correction rather than one-off fixes. This claim aligns with the debugging intuition that isolating failure-inducing differences enables systematic correction [6]. I study this claim in a production-representative financial Text-to-SQL setting under strict data-access constraints (Section 3).

## 2    Background and Related Work

*Enterprise-scale schema linking shifts the bottleneck upstream.* Benchmark progress shows that strong models can produce complex SQL when the target schema is bounded (e.g., Spider) [5]. Enterprise deployments change the regime: the difficulty often becomes filtering large schema pools and grounding terms to the correct subset of tables and columns under redundancy. LinkAlign frames scalable schema linking as database retrieval/schema pool filtering plus schema item grounding in large multi-database environments [4].

*Multi-step pipelines amplify intermediate errors without control surfaces.* When Text-to-SQL is embedded in multi-stage agent pipelines, incorrect intermediate outputs can propagate forward and become unchallengeable assumptions downstream. GUARDIAN highlights error injection and amplification in multi-agent collaboration, motivating engineered control surfaces for verification and correction [7].

*Why artifact-level localisation is an AI4SE mechanism.* Delta debugging formalises localising failure-inducing differences via systematic comparison [6]. I adopt the same diagnostic stance for Text-to-SQL by comparing intermediate commitments (grounded columns, selected tables, join paths, filters) against administrator-corrected references. This complements emerging views that LLM applications should be treated as systems requiring explicit testing protocols rather than only aggregate accuracy [1].

## 3    Industrial Case and Problem Definition

*Setting and boundaries.* I study an enterprise Text-to-SQL workload from a financial organisation under strict operational constraints. The study uses a production-representative *metadata snapshot* (900 tables, 22,000 columns, 24,000 domain terms, 3,000-word lexicon) and 69 anonymised business queries for paired

replay. Customer records and transaction data are excluded; the evaluation concerns schema/terminology grounding and SQL structure. This boundary is deliberate: enterprise Text-to-SQL difficulty is often dominated by schema pool filtering and grounding under redundancy and ambiguity [4].

*What "production-representative" means here.* Schemas and terminology reflect operational naming conventions and overlaps, and reference SQL is derived from administrator-approved or routinely executed queries. Because direct online execution and data access are restricted, I evaluate via offline replay on a frozen snapshot to enable reproducible paired comparisons.

*Baseline vs. revised agent (high-level).* The *baseline* follows a retrieval-and-generation pattern: retrieve schema evidence and generate SQL end-to-end. The *revised* configuration fixes major intermediate decisions as artifacts (grounding, table selection, join plan, filters) and compares them against references derived from corrected SQL. I focus on the baseline-versus-revised contrast because the contribution is the debugging mechanism rather than a sequence of model variants.

*Problem definition.* Enterprise failures frequently remain *silent*: SQL executes but answers the wrong intent due to mis-grounded terms, incorrect anchoring, meaningless joins, or ungrounded filters. The objective is to make these failures testable as software: localise the first wrong commitment and enable targeted correction.

## 4   Method: Artifact-Level Failure Localization

I localise failures by comparing generated intermediate artifacts against gold-derived reference artifacts, identifying the first wrong commitment (grounding, linking, join, or filter).

Figure 1 summarises the end-to-end process used in this paper: generate stage-wise artifacts, reconstruct gold-derived reference artifacts, and localise the first wrong commitment via deterministic comparison.

### 4.1   Pipeline commitments as artifacts

*Artifact schema.* I fix intermediate commitments as JSON artifacts emitted after major stages: (i) grounding artifact $\mathcal{A}_g$ (committed physical columns and optional value bindings), (ii) linking artifact $\mathcal{A}_t$ (selected tables, root/anchor, column-to-table bindings), (iii) join-plan artifact $\mathcal{A}_j$ (join edges and keys under admissibility rules), and (iv) filter artifact $\mathcal{A}_f$ (WHERE predicates and optional ORDER/LIMIT decisions). Downstream stages consume artifacts rather than free-form traces: $\mathcal{A}_g \rightarrow \mathcal{A}_t \rightarrow \mathcal{A}_j \rightarrow \mathcal{A}_f \rightarrow SQL$.

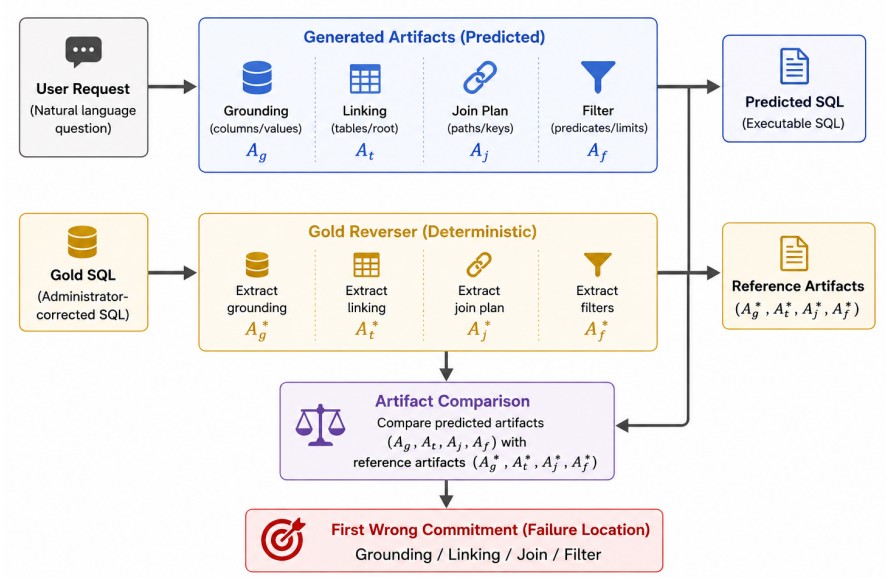

**Fig. 1.** Artifact-level failure localization: compare generated artifacts with gold-derived reference artifacts to label the first wrong commitment (grounding/linking/join/filter).

### 4.2  Gold SQL to reference artifacts (Gold Reverser)

*Gold SQL and Gold Reverser.* Gold SQL denotes administrator-approved (or production-derived) SQL expressing the intended meaning under the same snapshot. The deterministic *Gold Reverser* parses each Gold SQL into reference artifacts: $\mathcal{A}_g^\star$ (columns and literals), $\mathcal{A}_t^\star$ (tables and candidate root), $\mathcal{A}_j^\star$ (join edges/keys), and $\mathcal{A}_f^\star$ (predicates and optional ordering/limits). If production constructs are unavailable offline (e.g., views/parameters), I apply minimal adaptation while preserving intent.

*Why artifact matching instead of SQL-string matching.* Enterprise SQL often uses bridge tables for connectivity; strict string or set matching can penalise necessary structures. I therefore compare artifacts while separately reporting structural validity, consistent with enterprise linking constraints [4].

### 4.3  Divergence rules: localising the first wrong commitment

*What the paired replay asserts.* The 69-query paired replay evaluates whether the revised configuration reduces recurring grounding/structural failures under the same frozen snapshot. It does not claim universal Text-to-SQL superiority; it evaluates operational correctability.

**Table 1.** Example A: first divergence in the grounding artifact ($\mathcal{A}_g$).

| Item | Predicted ($\mathcal{A}_g$) | Reference ($\mathcal{A}_g^\star$) |
|---|---|---|
| Committed column for "status" | `status_history_code` | `current_status_code` |
| Evidence/source (example) | lexical partial match on `status` | admin-corrected intent alignment |
| Impact | semantic drift with valid execution | intent-preserving grounding |

*Deterministic divergence classification.* Given predicted artifacts ($\mathcal{A}_g, \mathcal{A}_t, \mathcal{A}_j, \mathcal{A}_f$) and references ($\mathcal{A}_g^\star, \mathcal{A}_t^\star, \mathcal{A}_j^\star, \mathcal{A}_f^\star$), I label the first wrong commitment in this order: (1) grounding failure (missing required columns, non-existent columns, or value bindings not grounded), (2) linking failure (wrong root/anchor or invalid bindings), (3) join violation (disconnected join islands or *inadmissible* join edges/keys), (4) filter invention/violation (unsupported predicates or ORDER/LIMIT without evidence). Here, *inadmissible* join edges/keys are those not permitted by the frozen join graph (or declared join-key constraints) used for the snapshot. The ordering prevents blaming downstream stages when upstream commitments already diverged.

## 5   Running Examples

I provide two anonymised examples that make artifact divergence visible. Each example shows the user request, a compact SQL fragment, and the artifact-level divergence label.

### 5.1   Example A: Semantic grounding failure (term drift)

*User request (anonymised).* *[REQ-A]* "List the current status of `[ENTITY]` accounts opened in `[PERIOD]`, excluding closed ones."

*Administrator-corrected SQL (fragment).*

```
SELECT ..., a.current_status_code
FROM ...
WHERE a.current_status_code <> 'CLOSED' ...
```

*Baseline divergence.* The baseline grounds "status" to a historical/derived field (`status_history_code`), yielding an executable but semantically drifted query. The first wrong commitment is therefore a grounding failure.

### 5.2   Example B: Structural failure (wrong anchor / meaningless join)

*User request (anonymised).* *[REQ-B]* "For `[ENTITY]` customers, show `[MEASURE]` by `[CATEGORY]` for `[PERIOD]`."

**Table 2.** Example B: divergence in root selection ($\mathcal{A}_t$) and join admissibility ($\mathcal{A}_j$).

| Signal | Predicted artifacts | Reference artifacts |
|---|---|---|
| Root/anchor ($\mathcal{A}_t$) | `Product` as root; bindings skew to product attributes | `Customer` as root; bindings cover customer/account identifiers |
| Join edges ($\mathcal{A}_j$) | includes `Customer.name = Product.owner_name` (name-based) | uses key-based joins (e.g., `customer_id`, `account_id`) |
| Violation label | meaningless join / weak key evidence | admissible join path |

*Administrator-corrected SQL (fragment).*

```
SELECT ...
FROM Customer c
JOIN Account a ON a.customer_id = c.customer_id
JOIN Fact f ON f.account_id = a.account_id ...
```

*Baseline divergence.* The baseline selects an inappropriate anchor table and constructs a plausible but business-invalid join using name similarity (e.g., `customer_name`=`owner_name`). The first wrong commitment is localized at linking/join planning.

## 6    Evaluation

### 6.1    Dataset and protocol

I evaluate via paired replay on the same frozen snapshot and 69 anonymised queries. Each configuration is executed under identical evidence resources so differences reflect the mechanism rather than changing context. The protocol targets operational correctability rather than universal SOTA claims.

### 6.2    Metrics

I report three compact metrics. **CLA** measures Macro-F1 over exact matching of physical column names. **TLA** measures Macro-F1 over exact matching of physical table names; in enterprise SQL, bridge tables may be necessary, so strict set-based TLA is conservative [4]. **SER** measures the rate of structural failures (parsing failure, catalog violation, or join-structure violation); execution-aware robustness motivates structural checks [3].

### 6.3    Results and interpretation

Table 3 summarises baseline vs. revised (N=69) under paired replay on the frozen snapshot.

**Table 3.** Paired replay results (N=69). Baseline vs. revised.

| Metric | Baseline | Revised |
|---|---|---|
| CLA (Macro-F1) ↑ | 0.543 | 0.782 |
| TLA (Macro-F1) ↑ | 0.327 | 0.547 |
| SER ↓ | 0.725 | 0.536 |

Higher CLA/TLA indicates more correct schema commitments, and lower SER indicates fewer structurally unsafe outputs. The actionable point for AI4SE is that artifact comparison improves diagnosability (localising the first wrong commitment) and supports cumulative correction [6, 1].

## 7    Discussion and Conclusion

*Lessons for AI4SE.* (1) Treat generated SQL as a software artifact: require traceability, regression suites, and structural checks. (2) Debug the first wrong commitment, not only final SQL. (3) Convert administrator corrections into reusable correction signals via gold-derived reference artifacts.

*Threats to validity.* This study covers one financial domain and 69 queries. Results depend on metadata quality (catalog/join constraints). Strict table-set matching may penalise necessary bridge tables. Offline replay may require minimal adaptation of production SQL constructs.

*Conclusion.* I contribute an operational debugging and correction loop for enterprise Text-to-SQL: intermediate commitments are fixed as artifacts, corrected SQL is transformed into reference artifacts, and deterministic comparison localises the first wrong commitment. Under paired replay, this improves diagnosability and supports cumulative reduction of recurring failures.

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
