# OpenReview forum: "Failure Localization for Enterprise Text-to-SQL Agents: Lessons from a Production-Representative Financial Case Study"
_KI/2026/Workshop/AI4SE — AI4SE Workshop_

### Official Review · Reviewer_LbP3 · 2026-06-12
**Fine-grain benchmarking Text2SQL Systems using gold-standard SQL**

**Rating:** 8
**Confidence:** 4

**Review:**

The paper presents an evaluation of Text2SQL models in a practical business environment, and promotes the possibility of correcting intermediate reasoning results of the Text2SQL pipeline with user-generated gold standard ground truth were available.

Strengths:
- I found the paper a very pleasant read, and the contribution is made (mostly) clear.
- The topic is certainly relevant and well within the scope of AI4SE, addressing the testing of AI models in database query pipelines.

Weaknesses:
- The paper lacks technical detail (specifically, regarding the specific Text2SQL model used, and a discussion of the results).
- Given the fact that the approach requires gold standard SQL for test queries, the paper should position its practical relevance clearer. To me, it became clear only in the final discussion that the paper focuses rather on software *testing*/benchmarking (given a test suite of queries with gold standard ground truth) than about improving model accuracy.

A few suggestions for improvement:
* Please provide more details about the actual Text2SQL models used. Particularly, how does their reasoning work (open reasoning loops vs guided workflows)? Are there prompts, and how do they look like? Have these models been fine-tuned specifically for Text2SQL, or are they based on instruction-tuned vanilla LLMs?
* Some more detail about the difficulty of the 69 test cases would be appreciated. For example, how many joins do they contain? How many tables/columns are affected on average?
* I assume that two SQL queries with very different structure can both be correct for the same request. How do you deal with this fact?
* Table 3 lacks a discussion. Some questions that came to my mind: 1. Why not report results by stage (grounding, linking, plan, filter)? After all, the possibility to identify bottleneck steps is promoted as a selling point of the paper. 2. Why isn't global correctness reported, i.e. in how many systems was the resulting SQL correct end-to-end? 3. Results seem quite poor overall. A critical discussion if the systems (given the risk of silent failures) have reached a stage of practical applicability would be appreciated. 4. I found the results for "revised" quite low. Doesn't this mean that each step is corrected with gold-standard SQL if needed? IMHO, this needs clarification.
* In Table 2, please separate the columns more clearly.

Overall, I found the paper interesting and relevant, and I believe it would make a useful contribution to the workshop.

---

### Official Review · Reviewer_P85E · 2026-06-12
**Artifact-level failure localization for enterprise Text-to-SQL: promising diagnostic framing, insufficient methodological detail**

**Rating:** 7
**Confidence:** 4

**Review:**

## Summary
The paper proposes an artifact-level failure localization method for LLM-based Text-to-SQL systems. The proposed approach introduces a debugging methodology that records intermediate decisions during the generation process (grounding, table linking, join planning, and filtering) as structured artifacts. These artifacts are compared against reference artifacts derived from administrator-corrected SQL to label the first incorrect commitment, enabling targeted correction. The approach is evaluated on 69 anonymized business queries against a production-representative financial metadata snapshot, showing improved schema-linking metrics and reduced structural errors compared with a baseline RAG pipeline.

## Strengths
- The paper addresses a relevant problem - silent failures in enterprise Text-to-SQL - and the framing as an AI4SE debugging problem, analogous to delta debugging is interesting and well motivated.
- The 'gold reverser' concept is novel (as far as i know) and a smart way to use existing organizational knowledge in a structured way without additional manual annotation

## Weaknesses
- The paper compares two approaches 'baseline' and 'revised', which are both described at high-level and lack methodological detail. For the revised method, I understand that the artifacts are compared against the gold standard, but I could not figure out what happens next, do you perform a correction loop? For both methods, there is no information regarding the LLM, retrieval method or prompting strategy, so it is 1) difficult to interpret the results in Table 3 and 2) impossible to reproduce or build on this work.
- The paper presents end-to-end evaluation results. Additional ablation experiments, evaluating the accuracy of the failure localization step alone would be interesting.
- The implementation of the 'gold reverser' could be explained in more detail. Also, its correctness is assumed but unvalidated, what happens if it misparses production SQL?

## Comments for the author
- I wonder how the method could be used in a practical setting. What would the workflow look like, If no matching reference artifacts could be found?
- Tables 1 and 2 are hard to read